# Towards the First Biomarker Test for Bipolar Spectrum Disorder: An Evaluation of 199 Patients in an Outpatient Setting

**DOI:** 10.3390/jpm13071108

**Published:** 2023-07-07

**Authors:** Andy Zamar, Ashma Mohamed, Abbi Lulsegged, Daniel Stahl, Christos Kouimtsidis

**Affiliations:** 1The London Psychiatry Centre, London W1G 7HG, UK; andy.zamar@psychiatrycentre.co.uk (A.Z.); ashmamohamed@gmail.com (A.M.); 2Endocrinologist Health 121 Ltd., London W1G 8QR, UK; admin@health121.co.uk; 3Department of Biostatistics and Health Informatics, Institute of Psychiatry, Psychology and Neuroscience, Kings College London, London SE5 8AF, UK; 1aniel.r.stahl@kcl.ac.uk; 4Department of Brain Sciences, Imperial College London, London W12 0NN, UK

**Keywords:** bipolar disorder, genetic testing, diagnosis

## Abstract

Bipolar spectrum disorder seems to be challenging to diagnose, particularly unspecified or subthreshold types. The delay in diagnosis in the UK for bipolar I and II types is a staggering 10–13 years, with only 15% correctly diagnosed without delay. In the USA, the delay is 6–8 years, and there is a 60% incorrect diagnosis rate. The HCL-32 questionnaire is adequate, but not sufficient by itself, and patients may find it difficult to complete, particularly if they are unwell. We have investigated a biomarker test which can be used in day-to-day clinical practice to assist diagnosis. We evaluated 199 patients diagnosed with ICD-10 bipolar I, II, and unspecified disorders, using the HCL-32 questionnaire with a cut-off point of 14 and above, supplemented by history taking and examination using the principles of the CIDI 3, interviews of relatives, and longitudinal mood charts where available. The results were compared to the general population and a sample of patients diagnosed with recurrent depression for assessment of sensitivity and specificity. We evaluated four mutations of SLCO1C1, DiO1, and two DiO2alleles as potential biomarkers for bipolar spectrum disorder, and identified three mutations that exhibited high sensitivity, with rates of up to 87% and specificity of up to 46% in distinguishing bipolar spectrum disorders from recurrent depressive disorder. Additionally, mutations in SLCO1C1 and DiO1 exhibited a sensitivity of up to 86% and a specificity of up to 60% in detecting bipolar spectrum disorder compared to the general population within a clinical setting. These biomarkers have the potential to be used as a diagnostic test that is not open to subjective interpretation and can be administered even if patients are very unwell, requiring, though, the patient’s consent. Further studies confirming these results are needed to compare the validity of using individual or a best combination of single nucleotide polymorphisms to identify bipolar spectrum disorders, particularly subthreshold presentations, and to differentiate them from other mood disorders such as major depression and recurrent depressive disorder.

## 1. Introduction

Bipolar disorder (BD) is a severe and enduring mental illness, and a highly heritable condition. According to the Global Burden of Disease study, BD is one of the most prevalent disabling health conditions in years lived with disability [1], due to its early onset [2] and delays in its diagnosis [3]. In 2019, 40 million people experienced BD [4]. These people have notably higher mortality rates, largely due to cardiovascular disease 38% and secondly due to suicide 18%, compared to the general population [5]. BD is a highly heritable disorder, with genetic influences explaining 60–85% of the risk through a review of family, twin, and adoption studies [6]. The diagnosis of BD often relies on the recognition of manic or hypomanic symptoms, which can be difficult to identify and distinguish from other psychiatric disorders, such as major depressive disorder or ADHD, leading to the potential for misdiagnosis [7]. It also leads to undiagnosed subthreshold presentations, and he consequences of inappropriate treatment. There is no laboratory diagnostic test for BD. A reliable diagnostic test could facilitate the early detection and identification of subthreshold presentations of the condition, elucidate underlying mechanisms, and accelerate the development of new personalised treatments. 

BD exists along a continuum, with varying degrees of severity and symptomatology recognised as bipolar spectrum disorder (BSD) [8]. BSD is based on the recognition that most cases of bipolar do not fit neatly into the traditional or meet the threshold diagnostic categories of bipolar I and II. In addition, both BD I and II present with inter-episode subthreshold symptoms, which are resistant to standard treatments and associated with polypharmacy [9]. BSD includes bipolar I, bipolar II, bipolar unspecified (ICD10 F31.9), and not otherwise specified (NOS) DSM 5 296.80, also known as subthreshold BD, which refers to cases in which individuals have some symptoms of BD but do not meet the full criteria for BD I or II of ICD 10. A notable study across 11 countries in 4 continents with 61,392 participants provided evidence for the validity of the concept of BSD, highlighting that the aggregate lifetime prevalence of BSD was 2.4%, with 0.6% for bipolar I, 0.4% for bipolar II, and 1.4% for subthreshold BD [8]. The lifetime prevalence of subthreshold BD is comfortably 40% more than both bipolar I and II put together, and is equally disabling, making it the most common subtype of BSD. 

The mean delay between illness onset and diagnosis is 5–13 years [4]. Delayed diagnosis of BD has significant consequences, including misdiagnosis, inappropriate treatment, increased morbidity, heightened suicide risk, strained relationships, impaired work and education performance, economic burden, and increased healthcare utilization, as well as early cardiovascular disease from the age of 18 [10]. 

Diagnosis of BSD is primarily based on clinical interviews, collateral history, and mood charts. The Hypomania Checklist-32 (HCL-32) is a self-report questionnaire designed to help screen for BD, specifically hypomanic symptoms. It was developed to address the need for a simple, easy-to-use tool for identifying hypomanic symptoms in clinical and research settings. The HCL-32 is not a definitive diagnostic tool, but rather a screening instrument that can help guide further clinical evaluation. In the original validation study of the HCL-32, its sensitivity was 80.0% and its specificity 51.4%. A cut-off score of 14 or higher indicates a positive screening result for BD [11]. 

Genetic epidemiological research has produced overwhelming evidence that genes affect predisposition to BSD. A recent GWAS study of approximately 42,000 patients with BD identified 15 genes robustly linked to BD-encoding druggable targets [12]. In another GWAS study including 20,352 cases of European descent, the findings were inconclusive; the results were consistent with small effect sizes and limited power, but also with genetic heterogeneity [13]. Several initiatives for candidate biomarkers in psychiatry have been undertaken in mood disorders in an attempt to identify, diagnose, and distinguish mood disorders, and a number of potential candidates have been identified for future potential validation. Currently, the situation suggests that the gains have overall been limited due to the heterogeneity of mood disorders, the limited clinical use of identified markers, and a limited understanding of how biomarkers affect pathophysiological states in mood disorders [14]. It was suggested that the probability of discovering diagnostic biomarkers that map precisely to specific disorders as defined in current classification systems is very low [15].

Mutations affecting the synthesis of deiodinase 1, 2 (DiO1 and DiO2), and SCL01C1 have been proposed to occur in BSD. Genetic studies have shown that polymorphism of the DiO2 gene (rs225014; Thr92Ala) is associated with depression, and heterozygote polymorphism has been associated with an increased risk of BD. Homozygous polymorphism of the DiO2 gene is associated with a 3.75-fold higher risk of BD [16]. By contrast, in recurrent depressive disorder, only 7% show homozygous mutations of DiO2, suggesting that the lower the mutations in DiO2 SNPs, the higher the risk of recurrent depressive disorder, a trend which opposes that observed in BSD [17]. It appears there is an optimal balance of DiO2 mutations within the general population, where too few mutations might precipitate recurrent depressive disorder, while too many could lead to BSD. Mutations of the 3′ UTR variant of DiO1 (rs11206244) have been associated with altered free thyroxine (FT4) levels in both White and African American subjects, as well as with lifetime major depressive disorder in White female subjects, in particular those from high-risk cohorts [17].

In prior cohort studies conducted by the current authors [18,19], a state of “cerebral hypothyroidism” was suggested due to DiO1, DiO2, and/or SLCO1C1 protein transporter-encoding gene mutations, which regulate levels of thyroid hormones in the brain tissues in BD. 

In this study, we concentrated on a small group of promising candidates to facilitate conclusive testing for the specific condition of BSD. Our methodology has primarily been grounded in clinical aspects, as opposed to a purely genetic focus. The approach evaluates pharmacodynamics and pharmacokinetics, and accidentally identified a potential test which needs further evaluation. In this study, our aim was to investigate the potential association between four specific genetic mutations (a variant of DiO1, two variants of DiO2, and SLCO1C1) and the risk of BSD, which led to the proposed development of the first potential laboratory-based test, and as such may help in early diagnosis and intervention. We accomplished this by comparing the frequencies of these mutations in a clinical population with their published prevalence rates in the general population, as well as with their occurrence in a recurrent depressive disorder population from Poland. This is particularly important for most patients with BD, namely unspecified BD, and when diagnostic clarity is uncertain in cases of subthreshold presentations. Identifying susceptibility genes will additionally pinpoint the biochemical pathways involved in pathogenesis, facilitate the development of more effective, better-targeted treatments, and offer opportunities for improving the validity of psychiatric diagnosis and classification.

## 2. Method

We conducted a genetic mutation analysis on a sample of 199 patients who met the ICD 10 criteria for type I, II and NOS BD. The results were compared with a population of between approximately 225,000 individuals for DiO1 rs2235544 mutations, 357,000 individuals for DiO2 rs220514 mutations, 299,000 individuals for DiO2 rs12885300 mutations, and 140,000 individuals for SLCO1C1 mutations within the general population. We also compared our results to the results of a cohort of 179 patients with recurrent depressive disorder diagnosed in a similar fashion with the Composite International Diagnostic Interview (CIDI) and ICD 10, and 152 healthy control subjects in relation to the above DiO2 SNPs [17]. The general population genomic data was obtained from the National Library of Medicine National Centre for Biotechnology Information [20], representing the general population through random sampling.

The clinical cohort was obtained from patients attending a private outpatient psychiatric clinic treated by five psychiatrists. Patients consented to receive genetic testing to assess suitability to start or continue high-dose thyroxine (HDT) treatment [18,19], as part of the UK Maudsley guidelines [9] for predicting tolerability and rationale for continued use or initiation of HDT. Patients had normal thyroid function tests upon presentation. Diagnosis of BSD was based on initial screening with the HCL-32 questionnaire (with a cut-off point of 14 and above), clinical history and examination using the principles of the CIDI 3 [21], longitudinal mood charts (where available), and collateral history from relatives (where possible). Ethical approval was not required, as the above interventions formed part of the standard clinical care outlined by the HDT treatment protocol provided by the treatment centre (training, accreditation, and licensing are available from the authors upon request). 

Genotyping for these mutations was performed using polymerase chain reaction (PCR) and subsequent DNA sequencing. The presence or absence of each mutation was determined for all participants. Buccal swabs were collected from all participants for DNA analysis. The Lifecode Gx™ panel for the central nervous system was examined, which includes testing for various neurotransmitters such as dopaminergic, adrenergic, and serotoninergic. Additionally, we specifically requested panel tests for several genetic variants, including the rs2235544 variant of the DiO1 gene, two variants of the DiO2 gene known as rs12885300 Gly3Asp and rs225014 Thr92Ala, and the rs10770704 intron3C>T variant of the SLCO1C1 gene, to judge tolerability for HDT treatment combined with rTMS. Data were collected by AM, who works independently of the treatment centre, and analysed by DS, who is also independent of the treatment centre. Lifecode Gx is also an independent company. 

## 3. Statistical Analysis

The genotype frequencies of the four genetic mutations (at least one mutation) were compared between the BSD clinical group and the control group and with published data from a study of patients with depressive disorder from Poland (N = 179), [17]. Differences between two groups were formally tested with a difference in proportion test, including an estimate of the difference and its 95% confidence interval [22]. 

Using a classification cut-off of 0.5, we calculated the sensitivity of each mutation in correctly identifying individuals with a BSD diagnosis, and the specificity in accurately identifying individuals in the control group without BSD. We also calculated the positive and negative predictive value of each mutation to correctly identify a case or a control individual, assuming a BSD prevalence of 2.4% within the population. In addition, we compared DiO2 mutations in our sample to a sample with recurrent depressive disorder from Poland [17] to assess the sensitivity and specificity of DiO2 in differentiating depression from BSD. No prevalence data were available for DiO1 or SLCO1C1 mutations, preventing a comparison with a pure depression group for these specific SNPs. Additionally, only summary measures were available for the two external databases, precluding further predictive analyses.

## 4. Results

Table 1 shows demographics, and clinical and genetic characteristics for 199 patients. Among the 199 patients in the study, the majority had a diagnosis of subthreshold BD (91%), while a small percentage had a diagnosis of bipolar I (3%) and II (6%). The mean age of the clinical population at the time of the test was 35.3 years (SD = 14.7, range: 13 to 80, N = 195), with the majority being females (57%). The age of onset was 25.2 years (range: 8–71, N = 162). The duration of illness at time of assessment was 9.4 years (range: 0 to 48, N = 159). Some 78% of the patients were white, and 10% were of Asian origin. Genetic analyses were available for between 178 and 199 cases, and patients with all genes tested had at least one mutation among the eight alleles for the four genes. On average, these patients had 4.5 mutations across all alleles (range 1–7, N = 177). Some 77% of the patients had an HCL-32 of 14 or higher (mean score: 18.3 (range 0–34)).

Figure 1 shows the prevalence of the four mutations within the clinical and general population. There was no significant difference between males and females (all *p* values > 0.23). The prevalence of three out of four mutations (DiO1, DiO2 b and SL01C1) was significantly higher in the clinical population compared to the general population, with mean differences ranging between 21% (DiO1) to 49% (SLCO1C1). There were small differences in the prevalence between the two groups in the DiO2 Gly3Asp allele (*p* = 0.67). There were small or negligible associations observed between the occurrence of at least one mutation among the four mutations. Only the association between DiO2 Thr92Ala and DiO2 Gly3Asp showed statistical significance, with a small kappa value of 0.22 (*p* < 0.001), while all other associations had *p*-values greater than 0.12 and kappa values below 0.13. A comparison of the clinical BSD population with the published data of a group with depressive disorder from Poland shows that the group with depressive disorder has significantly fewer DiO2 Thr92Ala mutations than the general population (12% less, *p* < 0.001) and the BSD population (33.1% less, *p* < 0.001), as shown in Table 2.

Table 3 shows that the sensitivity among the four alleles in correctly identifying persons with BSD ranged from 64.3% for DiO2a to 86% for SLCO1C1, while specificity, which measures the ability to correctly identify persons without BSD, was overall lower, ranging from 34.4% for DiO2 Gly3Asp to 62.8% for DiO2Thr92Ala. Positive predictive values were low (<5%) due to the low prevalence of BSD, while negative predictive values were very high, at about 98% for all four mutations. The negative predictive power is important, as 99% of those who do not show mutations will be healthy. 

## 5. Discussion

This study provides a first preliminary evaluation of a laboratory-based diagnostic test utilizing a biomarker for BSD, showing promising results in overcoming existing diagnostic and treatment barriers. Further research is necessary to validate and refine this biomarker’s potential in clinical applications.

In our sample, subthreshold BD presentations were in 91% of all presentations, which is higher than the figure reported in the literature. Whilst subthreshold BD was found to be on average 60% of BSD presentations, variation between studies taking place in different countries shows significantly wider ranges. Lifetime prevalence of subthreshold BD within BSD ranges from 50% in New Zealand to 78% and 80% in Romania and India, respectively [8]. We are not aware of a similar study in the UK. A number of patients were diagnosed as bipolar II, but subthreshold presentation was predominant and resistant to treatment, as reported in the literature [7,9]. The high proportion of subthreshold BD in our sample could be related to the prevalence in the UK, or the type of patients resistant to treatment presenting to our outpatient clinic; this is because subthreshold BD is usually misdiagnosed as depression, and if correctly diagnosed, is unresponsive to established treatment protocols [23].

The polymorphism detected in the GWAS studies was compromised by small effect sizes, the lack of a disease model to explain the condition, an overlap with schizophrenia and major depressive disorder, and remains of limited clinical value. Our approach is based on clinical findings in a cohort of patients with BSD, as diagnosed using the gold standard including psychometrics, patient evaluation, and mood charting. Regardless of ethnicity, all the studies for HDT use in BSD have shown that as a population, this group responds and uniquely tolerates high doses of thyroid hormones. Studies from Germany [24], the United States [25], and in UK populations [18,19] have demonstrated the same results. This led the authors to check the deiodinase enzymes’ status and intracerebral protein transporters, in order to evaluate the responses to and tolerability of treatment prior to prescribing as a form of precision medicine. It is well established that deiodinase enzyme regulation could be influenced by several factors such as medications, stress, and the intactness of the pituitary–thyroid axis. Regardless of the factors that influence deiodinase activity, the genotype remains stable for detection via testing. There were no differences between male and female patients in our BSD sample, as opposed to previous findings concerning major depressive disorder [17]. This adds to the potential argument that the two disorders are genetically different, hence the validity of the proposed test in differentiating between the two disorders.

The current authors proposed a disease model using genetics, molecular biology, symptoms, and treatment perspective, which to date explain BSD with the exception of the psychotic symptoms of bipolar I. Overlap with schizophrenia markers may explain this aspect, but bipolar I constitutes 25% of BSD; psychosis is not a core feature of the disorder, but occurs in a proportion of these patients [8].

A comparison to two different general populations, at least in the DiO2, does not show much difference from normal controls, but a striking difference in the prevalence of homozygous mutations between depressive disorders and BSD. A study in Germany with patients with BSD [24] shows the same response and tolerance as the US population [25] and the UK population [18,19], suggesting that they share the same features of DiO1/DiO2/SLCO1C1. We assume that the German and Polish population would not vary substantially given the long-intertwined history and geographical proximity of people and borders between the two countries. 

Three out of the four assessed mutations showed a significantly higher prevalence among individuals with a diagnosis of BSD compared to the general population, with a high sensitivity in correctly identifying those with a diagnosis of BSD. The presented estimates of sensitivity are conservative, because it is assumed a small proportion of 2.40% of participants in the general population also have BSD. Each of the three biomarkers had a sensitivity of 0.8 or more, and they are not correlated (almost independent), meaning that sensitivity may be even higher if we could assess the combination of the three alleles, for example, using a multivariable logistic regression with all three mutations as predictors. This was not possible, as raw data were not available for the general population. However, due to the low prevalence of BSD, genetic testing may result in a high number of false positive results, as indicated by the low positive predictive values of all genes. 

Genetic testing using buccal swabs offers a practical and accessible approach for clinical settings, as it can be conducted even when patients are unable or unwilling to complete self-report measures (such as the HCL-32) due to illness severity, misunderstanding, disagreement with the test, language barriers, or doubting the test’s validity. Furthermore, the identification of specific genetic mutations through this method may provide valuable information to clinicians regarding patients’ responses to antidepressants, as well as potential tolerance to high-dose levothyroxine, a proposed treatment strategy within the existing guidelines that is rarely used. It is also important for the initiation of early treatment for bipolar disorder in general, with little risk of deterioration to patients. This is because bipolar disorders are known to deteriorate with antidepressants, whilst the use of lithium, quetiapine, lamotrigine, and other mood stabilisers used in bipolar disorders can be used in depressive disorders with little damage to the course of either condition [23]. This approach may enable more personalized and effective treatment plans for individuals with BSD. The authors demonstrated in two previous cohorts a very high level of remission with few if any side effects, hypothesising that we may rely on and enhance mitochondrial dysfunction treatment and neuroplasticity. Validation and refinement of this diagnostic tool using representative populations followed by a randomised control trial is required to test the effectiveness of the treatment protocol against placebo or treatment as usual. Additionally, further research is needed to elucidate the underlying mechanism of the therapeutic effect. 

One can confidently argue that the disease progresses through selective neuronal hypothyroidism due to genetic mutations affecting intraneuronal thyroid hormone concentrations through inefficient transport and/or activation of T4 to T3. In addition, the DiO1, DiO2 Thr92Ala, and SLCO1C1 mutation SNPs show a high sensitivity and specificity in differentiating depressive disorder from BSD. Only 7% of patients with depressive disorder show DiO2 homozygous mutations (CC), compared to 34% of normal controls and 53% of patients with BSD, showing there should be a balance in the number of mutations, with a smaller number of mutations resulting in depressive disorders, and a higher number of mutations (compared to controls) resulting in BSD. 

Variants of DiO1 are associated with abnormal thyroid function tests at baseline, including elevated levels of reverse T3 and an elevated rT3/T3 ratio. DiO1, however, is thought to play less of a role in cerebral production of T3 than DiO2, and therefore its overall contribution to the initial pathogenesis of BSD is hitherto not fully understood. We do, however, argue that the presence of a DiO1 variant might play a greater role in protecting the periphery from the effects of high-dose levothyroxine, which has been shown in many studies [16], including the cohorts treated by the authors [18,19]. These mutations explain the disease model and mitochondrial dysfunction, as well as the model the authors published for the disease [26]. 

There are several limitations in this study. This study was conducted with a retrospective outpatient cohort, which was compared to a pre-existing sample on a database. Factors influencing deiodinase enzyme regulation, such as stress and medications as well as neuropsychiatric phenotypes, were not assessed and controlled, which may have limited the comparison of the sample with the sample from Poland. The sample size was small, despite covering a multi-ethnic group of patients attending the treatment centre.

## 6. Conclusions

Here, we report preliminary findings that support the development of a prediction tool to diagnose BSD. Despite several limitations, there is substantial evidence that mutations may function as markers for BSD. However, it is important to acknowledge that our ability to discuss the accuracy of the tool is currently limited. In clinical rather than epidemiological settings, in which patients present with depressive symptoms alongside other comorbidities or other risk factors affecting their mood, genetic testing (especially in DiO2) may be useful as an additional tool for the early identification of patients with BSD. Our study shows a large effect and features a multi-ethnic population, proposing an explanation for the disease model, from pathology to treatment, across several countries. This test will overcome unacceptable delays in diagnosis, as well as irreparable harm; it presents no risks or complications to patients. It should be used alongside HCL-32 and CIDI 3. 

A wider multi-ethnic, multi-country, large, and controlled study comparing major depressive disorder and BSD should be conducted using existing biobanks; alternatively, a comprehensive representative cohort study may be conducted, taking into consideration the longitudinal course of both conditions to differentiate the two conditions as accurately as possible, as well as other conditions such as ADHD and PTSD, whose clinical pictures overlap significantly with BSD. 

## Figures and Tables

**Figure 1 jpm-13-01108-f001:**
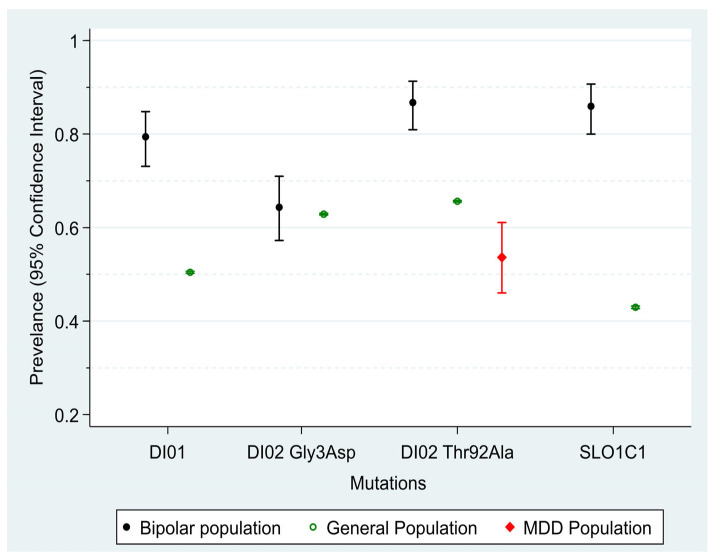
The prevalence of mutations (at least one allele) and their corresponding 95% confidence intervals for the bipolar, general population, and major depressive disorder (MDD) populations. The mean differences in prevalence between the bipolar and general population are as follows: DiO1: 29% (23.4% to 34.6%), *p* < 0.0001; DiO2 Gly3Asp: 1.5% (−5.2% to 8.1%), *p* = 0.67; DiO2 Thr92Ala: 21.1% (16.2% to 26.1%), *p* < 0.0001; and SLCO1C1: 43.0% (37.9% to 48.1%), *p* < 0.0001. The differences in DiO2 Thr92Ala prevalence between the bipolar population and the MDD population from Poland are 33.1% (24.3% to 41.9%), *p* < 0.0001, and between MDD and the general population are −12.0% (−19.3% to −4.7%), *p* = 0.0007. Further details are shown in Table 2.

**Table 1 jpm-13-01108-t001:** Demographics, clinical, and detailed genetic characteristics for all patients with bipolar. Proportions with 95% confidence intervals are reported. The sample size is 199 unless reported otherwise.

Variable	N	% (95% C.I.)
Sex (N = 198) Male	85	42.9% (36.2% to 49.9%)
Ethnicity		
White	156	78.4% (72.0% to 83.9%)
Black	5	2.5% (0.0% to 6.0%)
Mixed	2	1.0% (0.0% to 2.4%)
Asian	20	10.1% (6.3% to 15.1%)
Other/Middle Eastern	16	8.0% (4.7% to 12.7%)
Diagnosis (Ν = 198)		
BPAD subthreshold	181	90.9 (86.1% to 94.6%)
BPAD Type 1	6	3% (1.3% to 6.1%)
BPAD Type 2	12	6% (3.3% to 10%)
HCL-32 (179)	138	77.1 (70.0% to 83.0%)
Medical history		
PTDS/Trauma (N = 198)	99	50% (43.1% to 56.9%)
Substance use in the past	25	12.6% (8.5% to 17.7%)
Head injury	3	1.5% (0.4% to 4%)
Secondary diagnoses		
ADHD	39	19.6% (14.5% to 25.5%)
OCD		
Any eating disorder	7	3.5% (1.6% to 6.8%)
Current substance use	15	7.5% (4.5% to 11.8%)
Mutations		
DiO1 CC	41	20.6% (15.4% to 26.6%)
DiO1 AC	115	57.8% (50.9% to 64.5%)
DiO1 AA	43	21.6% (16.3% to 27.7%)
DiO2 a Gly3Asp TT	71	35.7% (29.3% to 42.5%)
DiO2 a Gly3Asp CT	95	47.7% (40.9% to 54.7%)
DiO2 a Gly3Asp CC	33	16.6% (11.9% to 22.2%)
DiO2 Thr92Ala TT (N = 181)	24	13.3% (8.9% to 18.8%)
DiO2 Thr92Ala CT (Ν = 181)	61	33.7% (27.1% to 40.8%)
DiO2 Thr92Ala CC (Ν = 181)	96	53% (45.8% to 60.2%)
SLC01C1 TT (N = 178)	25	14% (9.5% to 19.7%)
SLC01C1 CT (N = 178)	90	50.6% (43.3% to 57.8%)
SLC01C1 CC (N = 178)	63	35.4% (28.6% to 42.6%)
At least one of the four mutations(N = 177)	177	100% (97.9% to 100%)

**Table 2 jpm-13-01108-t002:** Prevalence of alleles (at least one) of the bipolar group, general population, and sample with depression from Poland.

Allele	Population N	Prevalence (95% C.I.)		Mean Difference (95% C.I.)	*p*
DiO1	BD (199)	79.4% (73.1% to 84.8%)	BD-GP:	29% (23.4% to 34.6%)	<0.0001
DiO1	GP (224,682)	50.4% (50.2% to 50.6%)			
DiO2 Gly3Asp	BD (199)	64.3% (57.2% to 71%)	BD-GP:	1.5% (−5.2% to 8.1%)	0.6712
DiO2 Gly3Asp	GP (357,390)	62.9% (62.7% to 63%)			
DiO2 Thr92Ala	BD (181)	86.7% (80.9% to 91.3%)	BD-GP:	21.1% (16.2% to 26.1%)	<0.0001
DiO2 Thr92Ala	GP (299,015)	65.6% (65.4% to 65.8%)			
DiO2 Thr92Ala	MDD (179)	53.6% (46.04 to 61.1%)	BD-MDD	33.1% (24.3% to 41.9%)	<0.0001
DiO2 Thr92Ala			MDD-GP	−12.0 (−19.3% to −4.7%)	0.0007
SLCO1C1	BD (178)	86% (80% to 90.7%)	BD-GP:	43% (37.9% to 48.1%)	<0.0001
SLCO1C1	GP (139,456)	43% (42.7% to 43.2%)			

BD = bipolar population, GP = general population, MDD = depression population from Poland, BD-GP = the mean difference between bipolar minus the general population, with a 95% confidence interval and a *p*-value test for difference in proportion. BD-POL and MDD-GB are similar comparisons. DiO1: rs2235544 variant of the DiO1 gene, DiO2 Gly3Asp = rs12885300 Gly3Asp variant, and DiO2 Thr92Ala = rs225014 Thr92Ala variant of the DiO2 gene, and SLCO1C1: rs10770704 intron3 C>T variant of the SLCO1C1 gene.

**Table 3 jpm-13-01108-t003:** Sensitivity, specificity, and positive and negative predictive values for alleles.

Gene	Sensitivity	Specificity	PPV	NPV
DIO1	79.40% (73.11% to 84.79%)	50.42% (50.21% to 50.62%)	3.71 (3.46% to 3.97%)	98.97% (98.66% to 99.21%)
DI02 Gly3Asp	64.32% (57.24% to 70.97%)	62.82% (62.66% to 62.98%)	4.08% (3.96% to 4.51%)	98.62% (98.34% to 98.85%)
DI02 Thr92Ala	86.74% (80.92% to 91.32%)	65.65% (65.48% to 65.82%)	3.15% (2.98% to 3.33%)	99.06% (98.64% to 99.35%)
SLCO1C1	85.96% (79.97% to 90.70%)	42.96% (42.70% to 43.22%)	3.57% (3.37% to 3.78%)	99.2% (98.86% to 99.44%)
^1^ DI02 Thr92Ala	86.74% (80.92% to 91.32%)	46.37% (38.90% to 53.96%)	N/A	N/A

Prevalence is used to calculate positive and negative predictive values. Confidence intervals for sensitivity and specificity are “exact” Clopper–Pearson confidence intervals. Confidence intervals for the predictive values are the standard logit confidence intervals. There is an assumed prevalence of 2.40%. ^1^ Comparison with depression group.

## Data Availability

Data is available at request form D.S.

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
