# Peer review of "Towards the First Biomarker Test for Bipolar Spectrum Disorder: An Evaluation of 199 Patients in an Outpatient Setting"

_jpm, 2023, doi:10.3390/jpm13071108_

Round 1

Reviewer 1 Report

This is a very interesting study assessing four SNPs from three genes involving thyroid hormones in 199 patients with BD, and also a sample of 179 patients with unipolar depression, and the general population. The authors show differences in prevalence of the previous mutations in the three populations and assess their capacity to be used as a diagnostic test with interesting results. 

Although the study is well written and provides interesting and applicable findings, some parts need to be revised: 

Major comments: 

The introduction is too long and detailed on clinical background. Only the last 2 paragraphs tackle genetics specifically. Please shorten it and focus on the topic of interest. 

Sample: From the 199 included patients, only 9% had BD1 or 2, and 91% Subthreshold BD. The high prevalence of BSD is unusual. Please discuss thoroughly along the manuscript since this will clearly limit the conclusions drawn. 

Discussion: You state that “sensitivity may be even higher if we could assess the combination of the three alleles. This was not possible as a combination was not available for the general population.” 

Please state how would you design a study to assess a predictive system considering all 4 mutations combined. 

Discussion: “We argue that this is evidence CEBM level 2b.” The sample size is low for a study in genetics and there is no follow-up. Please revise. 

Please include a “limitations” paragraph. 

Conclusion: “It should be used alongside the HCL-32 with a cut-off point of 14.”. How do you justify this statement? 

Minor comments: 

Please check: there are some abbreviations appearing as BP that should probably be BD along the manuscript (eg. both BP I and II present). 

“and patients with all genes tested had at least one mutation among the four alleles with an average of 4.5 mutations (range 1-7, N=177).” 

You mean 8 alleles for 4 genes? 

“There were little differences in the prevalence between the two groups in the DiO2 Thr92Ala allele (p=0.67).” 

If you check table 2 (and figure 1), are you referring to the Gly3Asp allele? The DiO2 Thr92Ala allele shows a mean diff of 21.1%. 

Figure 1: This figure is fantastic and would probably avoid the need of Table 2. Please include the prevalence of DiO2 Thr92Ala variant from the depression group from Poland. Please include the mean differences in the figure. 

Table 2: Please use BD instead of BP. Please use MDD or Dep instead of POL (Poland). Please merge the equal parts in column 1. 

Table 3 (prediction): Why the assumed prevalence you chose is 5%? 

Author Response

The would like to thank the reviewer for the very constructive comments. We attach our response to all comments made. 

This is a very interesting study assessing four SNPs from three genes involving thyroid hormones in 199 patients with BD, and also a sample of 179 patients with unipolar depression, and the general population. The authors show differences in prevalence of the previous mutations in the three populations and assess their capacity to be used as a diagnostic test with interesting results.

Although the study is well written and provides interesting and applicable findings, some parts need to be revised:

Major comments:

The introduction is too long and detailed on clinical background. Only the last 2 paragraphs tackle genetics specifically. Please shorten it and focus on the topic of interest.

Response: Thank you for the constructive advice. We have shortened the introduction and we have expanded the discussion and conclusion sections. We hope that the paper is now better balanced.

Sample: From the 199 included patients, only 9% had BD1 or 2, and 91% Subthreshold BD. The high prevalence of BSD is unusual. Please discuss thoroughly along the manuscript since this will clearly limit the conclusions drawn.

Response: The prevalence of Bipolar I and II within BSD is quite variant across different countries as reported by Merikangas et al. 2011. Furthermore, there are subthreshold presentations within the bipolar II as reported in the literature. We discuss these issues in the first paragraph of discussion.  We acknowledge that to our knowledge there is no data specific to the UK population.

Discussion: You state that “sensitivity may be even higher if we could assess the combination of the three alleles. This was not possible as a combination was not available for the general population.”

Please state how would you design a study to assess a predictive system considering all 4 mutations combined.

Response:  This could be done with a multivariable logistic regression with the three mutations as independent predictor variables. We added a sentence in the discussion. In addition, other potential predictors, such as HCL-32, could be included to improve the diagnostic accuracy. Alternatively, machine learning models could be employed to explore and analyse the predictive power of the combined mutations.

Discussion: “We argue that this is evidence CEBM level 2b.” The sample size is low for a study in genetics and there is no follow-up. Please revise.

Response: Thank you for your suggestion. We have deleted the sentence.

Please include a “limitations” paragraph.

Response: We have added the requested paragraph at the end of discussion.

Conclusion: “It should be used alongside the HCL-32 with a cut-off point of 14.”. How do you justify this statement?

Response:  As mentioned above, HCL-32 could increase the diagnostic power alongside the three mutations in a multivariable prediction model.  It will also increase acceptability among clinicians and patients who may be wary of prediction models.

Minor comments:

Please check: there are some abbreviations appearing as BP that should probably be BD along the manuscript (eg. both BP I and II present).

Response: We apologise for these mistakes. They are now corrected. 

“and patients with all genes tested had at least one mutation among the four alleles with an average of 4.5 mutations (range 1-7, N=177).” You mean 8 alleles for 4 genes?

Response: You are correct. We apologise for the mistake which is now corrected. 

“There were little differences in the prevalence between the two groups in the DiO2 Thr92Ala allele (p=0.67).” If you check table 2 (and figure 1), are you referring to the Gly3Asp allele? The DiO2 Thr92Ala allele shows a mean diff of 21.1%.

Response: We have corrected the mistake. Thank you.

Figure 1: This figure is fantastic and would probably avoid the need of Table 2. Please include the prevalence of DiO2 Thr92Ala variant from the depression group from Poland. Please include the mean differences in the figure.

Response:  We included the value from the polish group as requested. However, including the mean differences overloaded the figure, so we chose not to include it for the sake of clarity.

Table 2: Please use BD instead of BP. Please use MDD or Dep instead of POL (Poland). Please merge the equal parts in column 1.

Response: Modifications done. Thank you.

Table 3 (prediction): Why the assumed prevalence you chose is 5%?

Response: This was a mistake. 5% assumed prevalence refers to major depression not 2.40% for BSD as we state in the paper. We have corrected figures in table 3. 

Reviewer 2 Report

Manuscript number:      jpm-2414408

Title: Towards the first genetic laboratory test for Bipolar Spectrum Disorder: An evaluation of 199 patients in an outpatient setting

Reviewer Response

This manuscript is well written and represents an important addition to the current literature on the use and validity of biomarkers (individual vs combination of SNPS) in four genetic mutations (a variant of DiO1, two variants of DiO2, SLCO1C1) and the risk of bipolar spectrum disorder. The authors suggest the use of DiO2 may be useful as an additional tool along side behavior questionnaires to improve diagnosis between depressive disorders and bipolar spectrum disorders. This is an important area in psychiatry which also should be considered cautiously until findings can be replicated.

However, I have some minor concerns which may improve the manuscript:

Minor revisions

Background: The authors do an extensive introduction with the appropriate references. However, I believe it can be more succinct adding some context in the use of biomarkers in psychiatry (PMCID: PMC10168176)( PMCID: PMC10168147). Additionally, I believe that it should be stated previous studies examining the role of the deiodinase enzymes in different ethnic groups showing a higher association with major depression and females (Philibert et al., 2011). Another point of consideration is citing the different GWAS which have suggested different genetic variants with BD (Stahl, E.A., et al. Genome-wide association study identifies 30 loci associated with bipolar disorder. Nature genetics 51, 793-803 (2019)).

Results: All tables should be improved in terms of formatting and overall quality. Additionally, I would expect to see baseline scores for the HCL-32 as well as data on demographics of the sample (duration of last episode, age of onset of first depression etc).

Discussion: perhaps the authors should expand on the limitations on this study such as the study design, small sample size, the absence of considering ethnicity variations as well as on the overall longitudinal course of two distinct illnesses such as major depression and bipolar disorders.  It will be beneficial also adding that deiodinase enzyme regulation could be influenced by different medications, stress as well as the intactness of the pituitary–thyroid axis. Also, I believe the authors should expand on citing different studies on variations between gender and how that can be an influence.

Overall, the paper is well written and is clear in the presentation of their findings although these results as the authors point out need further replication and validity to be included as a method to distinguish BSD from major depressive disorders. 

Author Response

Reviewer 2

This manuscript is well written and represents an important addition to the current literature on the use and validity of biomarkers (individual vs combination of SNPS) in four genetic mutations (a variant of DiO1, two variants of DiO2, SLCO1C1) and the risk of bipolar spectrum disorder. The authors suggest the use of DiO2 may be useful as an additional tool alongside behavior questionnaires to improve diagnosis between depressive disorders and bipolar spectrum disorders. This is an important area in psychiatry which also should be considered cautiously until findings can be replicated.

However, I have some minor concerns which may improve the manuscript:

Minor revisions

Background: The authors do an extensive introduction with the appropriate references. However, I believe it can be more succinct adding some context in the use of biomarkers in psychiatry (PMCID: PMC10168176)( PMCID: PMC10168147). Additionally, I believe that it should be stated previous studies examining the role of the deiodinase enzymes in different ethnic groups showing a higher association with major depression and females (Philibert et al., 2011). Another point of consideration is citing the different GWAS which have suggested different genetic variants with BD (Stahl, E.A., et al. Genome-wide association study identifies 30 loci associated with bipolar disorder. Nature genetics 51, 793-803 (2019)).

Response: Thank you for your constructive suggestions. We have added the suggested additional references and we hope that the background section has improved to desired standards. 

Results: All tables should be improved in terms of formatting and overall quality. Additionally, I would expect to see baseline scores for the HCL-32 as well as data on demographics of the sample (duration of last episode, age of onset of first depression etc).

Response: We have re-formatted all tables. We hope it is easier to follow. We have added data on ethnicity, age of onset, duration of illness and HCL-32.

Discussion: perhaps the authors should expand on the limitations on this study such as the study design, small sample size, the absence of considering ethnicity variations as well as on the overall longitudinal course of two distinct illnesses such as major depression and bipolar disorders.  It will be beneficial also adding that deiodinase enzyme regulation could be influenced by different medications, stress as well as the intactness of the pituitary–thyroid axis. Also, I believe the authors should expand on citing different studies on variations between gender and how that can be an influence.

Response:

We have added a paragraph at the end of discussion section. We have discussed in extend the lack of difference between genders as well as ethnicity variations and their potential significance. In conclusion we have acknowledged that these are preliminary findings and proposed the need for a multi-ethnic, multi-country, longitudinal study that could provide robust support in the utility of the proposed diagnostic test. We hope that the points discussed are adequate.

Reviewer 3 Report

In this manuscript, Zamar and colleagues describe their approach to developing a genetic test for bipolar spectrum disorder (BSD) using four genetic elements in for genes. The idea of establishing laboratory diagnostic tests for BD is noteworthy, with important downstream implications for the patients and the medical field. However, as presented, due to severe methodological shortcomings (outlined below), this manuscript falls short of establishing such a laboratory test for predicting BD status. My comments are provided below in the order of my reading of the manuscript. 

1. My main concern pertains to the complete lack of a proper statistical framework to properly evaluate the significance of the chosen polymorphisms. If this reviewer is correct, the authors are attempting to identify disease-based genetic biomarkers for BD. Even though the authors talk about a laboratory test, they, in fact, have not developed (established) a laboratory test but attempted to provide a statistical assay to demonstrate that the selected polymorphisms could be used to identify BD patients from controls. Unfortunately, their approach is inadequate. There are many approaches to validate genetic biomarkers’ predictive capabilities, a simple and relatively robust approach would be to use a ROC curve analysis to assess the sensitivity and specificity of the selected genetic elements. However, none of this is presented here. 

2. My second comment concerns the lack of a proper demographic description of the selected subjects. This is important, especially when the authors are comparing the genetic burden (whatever they mean by that) in what I believe to be subjects from the UK vs. subjects from Poland. 

3. I am further confused about the selection criteria for polymorphisms in the selected genes. It would have made much more sense to select polymorphisms from the recent GWAS of BD rather than focusing on genes that are involved in thyroid metabolism. In fact, this raises additional concern, mainly because it is quite possible that the BD phenotype is comorbid to the main metabolic disease. 

4. The authors have not properly explained the purpose of comparing the results from a selected diagnostic group against a population-based sample. This is important if the subjects have not been screened for neuropsychiatric phenotypes. While comparing their results against a BD sample from Poland may be warranted the lack of proper demographic description, makes these comparisons difficult to impossible to evaluate.

None

Author Response

Thank you for your comments. We acknowledge that we might have confused reviewers. We hope that our revisions have clarified issues raised. 

In this manuscript, Zamar and colleagues describe their approach to developing a genetic test for bipolar spectrum disorder (BSD) using four genetic elements in for genes. The idea of establishing laboratory diagnostic tests for BD is noteworthy, with important downstream implications for the patients and the medical field. However, as presented, due to severe methodological shortcomings (outlined below), this manuscript falls short of establishing such a laboratory test for predicting BD status. My comments are provided below in the order of my reading of the manuscript.

  1. My main concern pertains to the complete lack of a proper statistical framework to properly evaluate the significance of the chosen polymorphisms. If this reviewer is correct, the authors are attempting to identify disease-based genetic biomarkers for BD. Even though the authors talk about a laboratory test, they, in fact, have not developed (established) a laboratory test but attempted to provide a statistical assay to demonstrate that the selected polymorphisms could be used to identify BD patients from controls. Unfortunately, their approach is inadequate. There are many approaches to validate genetic biomarkers’ predictive capabilities, a simple and relatively robust approach would be to use a ROC curve analysis to assess the sensitivity and specificity of the selected genetic elements. However, none of this is presented here.

Response

We thank the reviewer for this very important comment. This study does not put forth a predictive model for a biomarker; instead, it examines the potential of certain alleles as promising biomarkers, a proposition that warrants further exploration. In an initial attempt to appraise the potential, and acknowledge the limitations, of these alleles as biomarkers, we present both sensitivity and specificity calculations at a somewhat arbitrary threshold of 0.5. Because we do not have the raw data for the global and Polish studies to calculate the predicted probability, we are not able to present a ROC curve or a summary measure, such as the AUC for the model's capacity for distinguishing between classes.

We acknowledge that we might have potentially confused readers. To that effect we have explained extensively both in the last paragraph of the background section as well as in the first couple paragraphs of the discussion section the clinical rather than the laboratory or epidemiological ankle of our study. Given the observed great response to High Dose Levothyroxine treatment of our patients we wanted to investigate what if any genetic differences are existing between them, the general population and patients with major depressive disorder. We hope that the added clarifications add to the scientific value of the paper.

  1. My second comment concerns the lack of a proper demographic description of the selected subjects. This is important, especially when the authors are comparing the genetic burden (whatever they mean by that) in what I believe to be subjects from the UK vs. subjects from Poland.

Response

We have added information regarding the ethnicity of our sample. We have also discussed in the first few paragraphs of discussion the potential role of ethnicity to our findings. We did not collect data on the course of their condition such as age of onset of the condition and duration of the most recent episode. We list these as limitations to the study. I hope that this is acceptable.

  1. I am further confused about the selection criteria for polymorphisms in the selected genes. It would have made much more sense to select polymorphisms from the recent GWAS of BD rather than focusing on genes that are involved in thyroid metabolism. In fact, this raises additional concern, mainly because it is quite possible that the BD phenotype is comorbid to the main metabolic disease.

Response

We thank the reviewer for this important issue. Given the limited utility of the recent GWAS findings, which are discussed in the background section of the paper, we think we justify well our approach and the focus on a small number of genes associated with the deiodinase enzyme regulation for the following reasons: 1) we have observed in clinical practice the great response to High Dose Levothyroxine treatment without the experience of side effects, 2) existing literature suggesting potential link between this deiodinase enzyme system and mood disorders, and 3) availability of another study of a sample with BSD diagnosed in the same way investigating the same genes.

We have discussed in extend the proposed disease model of BSD. It is important to note that our sample had normal thyroid function test at presentation.  

  1. The authors have not properly explained the purpose of comparing the results from a selected diagnostic group against a population-based sample. This is important if the subjects have not been screened for neuropsychiatric phenotypes. While comparing their results against a BD sample from Poland may be warranted the lack of proper demographic description, makes these comparisons difficult to impossible to evaluate.

Response:

Thank you for this important comment. We have acknowledged that this paper presents preliminary findings. We also acknowledge that we did not control for other factors affecting the deiodinase enzyme regulation system nor for neuropsychiatric phenotypes. We have emphasised that the sample was clinical and the approach empirical. We hope that the added explanations are sufficient.

Round 2

Reviewer 1 Report

Thanks for addressing the previous comments. Congratulations on such a relevant work.

Reviewer 3 Report

I thank the reviewers for their responses.